# Structure of mycobacterial CIII$_2$CIV$_2$ respiratory supercomplex bound to the tuberculosis drug candidate telacebec (Q203)

David J Yanofsky[1,2†], Justin M Di Trani[1†], Sylwia Król[3], Rana Abdelaziz[4], Stephanie A Bueler[1], Peter Imming[4]*, Peter Brzezinski[3]*, John L Rubinstein[1,2,5]*

[1]Molecular Medicine Program, The Hospital for Sick Children, Toronto, Canada; [2]Department of Medical Biophysics, The University of Toronto, Toronto, Canada; [3]Department of Biochemistry and Biophysics, Stockholm University, Stockholm, Sweden; [4]Department of Pharmaceutical/Medicinal Chemistry and Clinical Pharmacy, Martin-Luther-Universitaet Halle-Wittenberg, Halle (Saale), Germany; [5]Department of Biochemistry, The University of Toronto, Toronto, Canada

*For correspondence:
peter.imming@pharmazie.uni-halle.de (PI);
peterb@dbb.su.se (PB);
john.rubinstein@utoronto.ca (JLR)

†These authors contributed equally to this work

**Abstract** The imidazopyridine telacebec, also known as Q203, is one of only a few new classes of compounds in more than 50 years with demonstrated antituberculosis activity in humans. Telacebec inhibits the mycobacterial respiratory supercomplex composed of complexes III and IV (CIII$_2$CIV$_2$). In mycobacterial electron transport chains, CIII$_2$CIV$_2$ replaces canonical CIII and CIV, transferring electrons from the intermediate carrier menaquinol to the final acceptor, molecular oxygen, while simultaneously transferring protons across the inner membrane to power ATP synthesis. We show that telacebec inhibits the menaquinol:oxygen oxidoreductase activity of purified *Mycobacterium smegmatis* CIII$_2$CIV$_2$ at concentrations similar to those needed to inhibit electron transfer in mycobacterial membranes and *Mycobacterium tuberculosis* growth in culture. We then used electron cryomicroscopy (cryoEM) to determine structures of CIII$_2$CIV$_2$ both in the presence and absence of telacebec. The structures suggest that telacebec prevents menaquinol oxidation by blocking two different menaquinol binding modes to prevent CIII$_2$CIV$_2$ activity.

## Introduction

Numerous bacteria from the strictly aerobic genus *Mycobacterium* are human pathogens. In particular, infection by *Mycobacterium tuberculosis* and closely related species results in the disease tuberculosis (TB). In most years, TB is the leading cause of death by infectious disease internationally, with an increasing incidence of drug-resistant infections (*Global Tuberculosis Report, 2020*). Nontuberculosis mycobacterial pathogens include *Mycobacterium leprae*, which causes leprosy, *Mycobacterium ulcerans*, which causes Buruli ulcer, and *Mycobacterium avium* and *Mycobacterium abscessus*, which infect immunocompromised and cystic fibrosis patients, respectively. The discovery of bedaquiline from a phenotypic screen with non-pathogenic *Mycobacterium smegmatis*, and its subsequent development into an effective therapeutic, has revolutionized the treatment of multidrug-resistant and extensively drug-resistant TB (*Global Tuberculosis Report, 2020*; *World Health Organization, 2019*). Bedaquiline binds the membrane region of mycobacterial ATP synthase (*Andries et al., 2005*; *Guo et al., 2021*; *Preiss et al., 2015*), blocking proton translocation and ATP synthesis. Thus, in addition to providing an invaluable therapeutic tool, bedaquiline established oxidative phosphorylation as a target space for antibiotics against mycobacteria. Subsequent to the discovery of bedaquiline,

numerous compounds have been identified that target either ATP synthase or the electron transport chain complexes that establish the transmembrane proton motive force (PMF) that drives ATP synthesis (*Cook et al., 2017*).

In mammalian mitochondrial electron transport chains, complexes I and II oxidize NADH (the reduced form of nicotinamide adenine dinucleotide) and succinate, respectively. The electrons from these substrates are used to reduce a membrane-bound pool of ubiquinone (UQ) to ubiquinol (UQH$_2$). Electrons from UQH$_2$ are then passed successively to CIII (also known as cytochrome [cyt.] $bc_1$), soluble cyt. *c*, and CIV (also known as cyt. *c* oxidase or cyt. $aa_3$) before ultimately reducing molecular oxygen to water. Complexes I, III, and IV couple electron transfer to proton transfer across the membrane, thereby generating the PMF that drives ATP synthesis. In contrast to mitochondria and many bacteria, mycobacteria possess a branched electron transport chain (reviewed in *Cook et al., 2017*). Rather than UQ, mycobacterial respiration relies on menaquinone (MQ) as an intermediate electron carrier. MQH$_2$ can reduce molecular oxygen via two MQH$_2$:O$_2$ oxidoreductases: cyt. *bd* and cyt. $bcc$-$aa_3$, the latter being equivalent to a combination of canonical CIII and CIV with the stoichiometry CIII$_2$CIV$_2$. The CIII$_2$CIV$_2$ supercomplex and cyt. *bd* branches of the mycobacterial electron transport chain are bioenergetically not equivalent, while CIII$_2$CIV$_2$ transfers three protons across the membrane for each electron used to reduce O$_2$, cyt. *bd* transfers the equivalent of only one proton across the membrane for each electron. Enzyme utilization in mycobacterial electron transport chains can adapt to changes in environmental conditions and treatment with respiratory complex inhibitors (*Arora et al., 2014*; *Berney and Cook, 2010*), which complicates targeting of respiration by antimycobacterial drugs (*Beites et al., 2019*).

Structural analysis of the CIII$_2$CIV$_2$ supercomplex from *M. smegmatis* led to the discovery that a dimeric type C superoxide dismutase (SOD) is an integral component of the assembly (*Gong et al., 2018*; *Wiseman et al., 2018*). The SOD dimer is found on the periplasmic side of CIII$_2$CIV$_2$ and is held in place by its two N-terminal tails, which bind to the complex's QcrA subunits. Both QcrA and the SOD subunits are highly conserved between *M. smegmatis* and *M. tuberculosis*, with 81.4 % and 71.0 % sequence similarity and 73.2% and 64.8% sequence identity, respectively, suggesting that a similar association occurs in the *M. tuberculosis* enzyme. *M. tuberculosis* mutants that lack SOD are susceptible to killing within macrophages (*Piddington et al., 2001*), emphasizing the importance of the subunit. Given its position within CIII$_2$CIV$_2$, it is possible that the SOD subunit abstracts electrons from reactive oxygen species formed during respiration or generated by host-defense mechanisms in the phagolysosome upon phagocytosis of *M. tuberculosis* and directs them through the respiratory chain to contribute to the PMF and ATP synthesis (*Gong et al., 2018*; *Wiseman et al., 2018*).

Although killing of *M. tuberculosis* with electron transport chain inhibitors may require simultaneously blocking both the CIII$_2$CIV$_2$ and cyt. *bd* branches for oxygen reduction (*Arora et al., 2014*; *Beites et al., 2019*; *Matsoso et al., 2005*), high-profile candidate TB therapeutics have been identified that bind to CIII within CIII$_2$CIV$_2$ (*Pethe et al., 2013*; *Rybniker et al., 2015*). Similarly, while CIII$_2$CIV$_2$ is not essential in *M. smegmatis*, its disruption causes severe growth defects (*Matsoso et al., 2005*). In contrast, CIII$_2$CIV$_2$ is essential in *M. leprae* and *M. ulcerans*, which lack the cyt. *bd* branch of the electron transport chain entirely (*Cole et al., 2001*; *Demangel et al., 2009*).

Rather than pumping protons, CIII contributes to the PMF by separating positive and negative charges across the membrane through a mechanism known as the Q cycle (reviewed in *Sarewicz and Osyczka, 2015*; *Xia et al., 2013*). Each CIII contains two sites where redox reactions with MQ occur: a Q$_P$ site near the positive (periplasmic) side of the membrane where MQH$_2$ is oxidized and a Q$_N$ site near the negative (cytoplasmic) side of the membrane where MQ is reduced. In the current understanding of the Q cycle in mycobacteria, oxidation of MQH$_2$ in the Q$_P$ site leads to release of two protons to the positive side of the membrane. The first electron from this oxidation reaction is passed to a [2Fe–2 S] Rieske center (FeS) in subunit QcrA where it consecutively reduces the cyt. *cc* domain of the QcrC subunit and the Cu$_A$ di-nuclear center of CIV. The second electron from the MQH$_2$ in the Q$_P$ site is transferred first to heme $b_L$ and then heme $b_H$, both in subunit QcrB of CIII, before reducing a MQ molecule in the Q$_N$ site to MQ$^{\bullet-}$. Oxidation of a second MQH$_2$ in the Q$_P$ site and repetition of this series of events lead to reduction of MQ$^{\bullet-}$ to MQH$_2$ in the Q$_N$ site, upon abstraction of two protons from the negative side of the membrane, thereby contributing to the PMF and the pool of reduced MQH$_2$ in the membrane. Within CIV, electrons are transferred from the Cu$_A$ di-nuclear center to O$_2$

bound at the heme $a_3$-$Cu_B$ binuclear catalytic site via heme $a$, driving proton pumping across the membrane.

Telacebec (also known as Q203) was identified in a screen of macrophages infected with *M. tuberculosis* (*Pethe et al., 2013*). Generation of resistance mutants bearing T313I and T313A mutations in the *qcrB* gene indicated that telacebec targets CIII of the $CIII_2CIV_2$ supercomplex. A recent Phase 2 clinical trial demonstrated a decrease in viable mycobacterial sputum load with increasing dose of telacebec, supporting further development and making telacebec one of only a few new drug classes in more than 50 years with demonstrated antituberculosis activity in humans (*de Jager et al., 2020*). Telacebec may also have clinical utility in treating nontuberculosis mycobacterial infections, such as Buruli ulcer (*Almeida et al., 2020*; *Van Der Werf et al., 2020*).

Here, we use electron cryomicroscopy (cryoEM) to investigate how telacebec inhibits purified $CIII_2CIV_2$ from *M. smegmatis*. We develop conditions for $CIII_2CIV_2$ activity assays that limit the spontaneous autoxidation of $MQH_2$ analogues, which has hampered previous analysis of $MQH_2$:$O_2$ oxidoreductase activity with purified $CIII_2CIV_2$. The assays show that telacebec inhibits $CIII_2CIV_2$ activity at concentrations comparable to those that inhibit electron transfer in mycobacterial membranes. CryoEM of $CIII_2CIV_2$ demonstrates both the presence of the LpqE subunit (*Gong et al., 2018*) and different conformations of the cyt. *cc* domain (*Wiseman et al., 2018*), which have previously been observed separately but not together. Three-dimensional variability analysis (3DVA) of the structure shows that the SOD subunit can move toward cyt. *cc*, supporting the possibility of direct electron transfer from superoxide to CIV. CryoEM of the $CIII_2CIV_2$:telacebec complex allows localization of the telacebec binding site with the imidazopyridine moiety and A-benzene ring of telacebec forming most protein-inhibitor contacts.

## Results

### Structure of $CIII_2CIV_2$ reveals movement of SOD subunit and cyt. *cc* domain

In order to facilitate isolation of $CIII_2CIV_2$, we used the ORBIT (oligonucleotide-mediated recombineering followed by Bxb1 integrase targeting) strategy (*Murphy et al., 2018*) to introduce sequence for a 3×FLAG affinity tag into the chromosomal DNA of *M. smegmatis* immediately 3′ to the *qcrB* gene. While *M. smegmatis* is typically grown in 7H9 medium supplemented with albumin, dextrose, and sodium chloride (ADS), we found that supplementing instead with tryptone, dextrose, and sodium chloride (TDS), which is more economical for large-scale culture, gave equivalent or superior growth. Purification of $CIII_2CIV_2$ from *M. smegmatis* grown in these conditions gave a high yield of enzyme with clear bands on an SDS-PAGE gelfor most of the known subunits of the complex (*Figure 1A*). We observed that following affinity purification, gel filtration chromatography of the enzyme led to depletion of the LpqE and SOD subunits (*Figure 1A*, *right*) compared to affinity purification alone (*Figure 1A*, *left*), and consequently this purification step was avoided.

CryoEM of the $CIII_2CIV_2$ preparation allowed for calculation of a 3D map of the enzyme at a nominal resolution of 3.0 Å (*Figure 1B*, *Figure 1—figure supplement 1*, and *Table 1*). The map shows strong density for the LpqE subunit (*Figure 1B*, *orange*). LpqE was observed in one previous structural study of $CIII_2CIV_2$ from *M. smegmatis* (*Gong et al., 2018*) but was absent in another (*Wiseman et al., 2018*), presumably due to depletion of the subunit during purification of the supercomplex. In the structure missing LpqE, the cyt. *cc* domain of subunit QcrC adopts both an 'open' and a 'closed' conformation, while the structure with LpqE was found only in the closed conformation. The closed conformation creates a direct electronic connection between heme $c_{II}$ of CIII and $Cu_A$ of CIV (*Gong et al., 2018*; *Wiseman et al., 2018*). In the open conformation, heme $c_{II}$ from the cyt. *cc* domain is too far from $Cu_A$ to allow electron transfer, leading to the hypothesis that switching between the closed and open conformations plays a role in controlling the flow of electrons through the supercomplex (*Wiseman et al., 2018*). In contrast, LpqE was hypothesized to strengthen the physical attachment between CIII and CIV (*Gong et al., 2018*). 3DVA with the current dataset (*Punjani and Fleet, 2021*), focused on one half of the supercomplex, revealed complexes with and without LpqE. Where LpqE was missing, the cyt. *cc* domain exhibits the open conformation, while complexes with LpqE show only the closed conformation of cyt. *cc* (*Figure 1—figure supplement 2*, *Video 1*). Clashes between LpqE and the open conformation of cyt. *cc* suggest that LpqE prevents the open conformation.

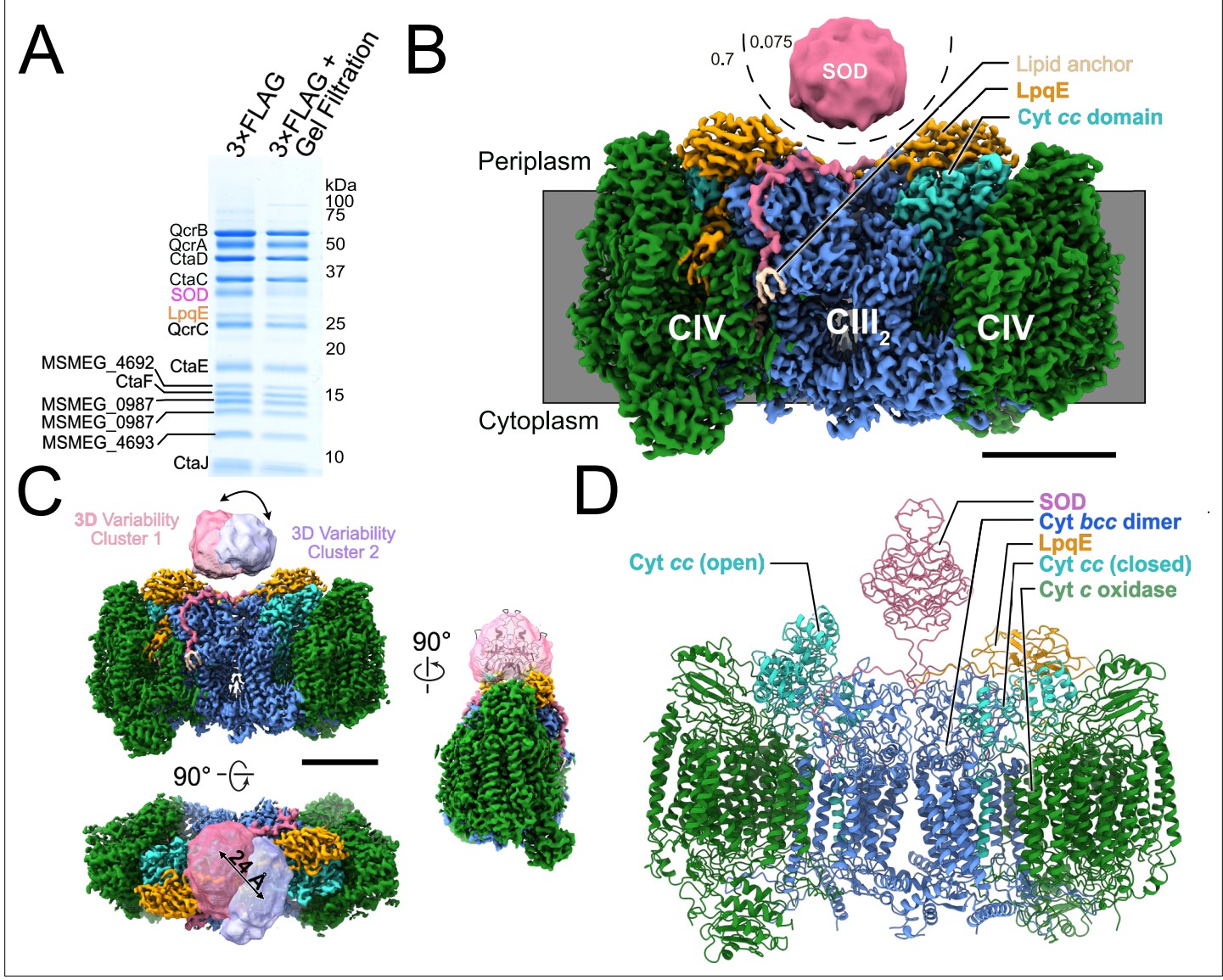

**Figure 1.** Structure of the *Mycobacterium smegmatis* CIII₂CIV₂ respiratory complex. (**A**) SDS-PAGE shows most of the known subunits of the complex and indicates that the superoxide dismutase (SOD) and LpqE subunits are depleted by gel filtration chromatography. (**B**) Electron cryomicroscopy (CryoEM) map of the CIII₂CIV₂. The different density thresholds for the SOD subunit and the rest of the complex are indicated. Scale bar, 50 Å. (**C**) Three-dimensional variability analysis indicates two different clusters of particle images ('cluster 1' and 'cluster 2') that show the SOD subunit in different positions over the twofold symmetry axis of the complex. Scale bar, 50 Å. (**D**) An atomic model for the CIII₂CIV₂ complex with SOD fitted into the map and showing one half of the complex missing the LpqE subunit and with the cyt. *cc* domain in the 'open' conformation and the other half of the complex possessing the LpqE subunit and with the cyt. *cc* domain in the 'closed' conformation.

The online version of this article includes the following figure supplement(s) for figure 1:

**Figure supplement 1.** Electron cryomicroscopy (CryoEM) map validation.

**Figure supplement 2.** Three-dimensional (3D) variability analysis of LpqE and the cyt. *cc* domain.

**Figure supplement 3.** Examples of model in map fit.

In previous studies, the SOD subunit of CIII₂CIV₂ was poorly resolved in cryoEM maps and appeared as a diffuse density (*Gong et al., 2018*; *Wiseman et al., 2018*). In the present map, the overall shape of SOD, although still at lower density than the rest of the complex, was more readily apparent (*Figure 1B*, *pink*). The N-terminal anchors from the SOD dimer that bind to subunit QcrB are well resolved and terminate at the middle of the complex with a lipid anchor (*Figure 1B*, *beige*). The improved density for the SOD subunit allowed fitting of a homology model of the protein into the

**Table 1.** Electron cryomicroscopy (CryoEM) structure determination.

**A. CryoEM data acquisition and image processing**

| Data collection | |
| --- | --- |
| Electron microscope | Titan Krios G3 |
| Camera | Falcon 4 |
| Voltage (kV) | 300 |
| Nominal magnification | 75,000 |
| Calibrated pixel size (Å) | 1.03 |
| Total exposure (e/Å²) | 43.5 |
| Exposure rate (e/pixel/s) | 5.99 |
| Number of exposure fractions | 29 |
| Defocus range (μm) | 0.7–2 |
| Image processing | |
| Motion correction software | *cryoSPARC v3* |
| CTF estimation software | *cryoSPARC v3* |
| Particle selection software | *cryoSPARC v3* |
| Micrographs used in inhibitor-free dataset | 4308 |
| Micrographs used in telacebec-bound dataset | 2793 |
| Particle images selected in inhibitor-free dataset | 1,037,709 |
| Particle images selected in telacebec-bound dataset | 387,777 |
| 3D map classification and refinement software | *cryoSPARC v3* |

**B. Model statistics**

| Dataset | Inhibitor-free | Telacebec-bound |
| --- | --- | --- |
| Associated PDB ID | | |
| Modeling and refinement software | *Coot, phenix, ISOLDE* | *Coot, phenix, ISOLDE* |
| Protein residues | 6058 | 6075 |
| Ligand | 9 XX: 4, 9Y0: 6, 9YF: 8, FES: 2, HEC: 4, HEA: 4, MQ9: 10, HEM: 4, PLM: 4, CU: 8 | 9 XX: 4, 9Y0: 6, 9YF: 8, FES: 2, HEC: 4, HEA: 4, MQ9: 10, HEM: 4, PLM: 4, CU: 8, QTE: 1 |
| RMSD bond length (Å) | 0.005 | 0.004 |
| RMSD bond angle (°) | 0.712 | 0.818 |
| Ramachandran outliers (%) | 0.22 | 0.2 |
| Ramachandran favored (%) | 92.58 | 91.52 |
| Rotamer outliers (%) | 0 | 0 |
| Clash score | 19.29 | 15.47 |
| MolProbabity score | 2.25 | 2.20 |
| EMRinger score | 3.12 | 2.61 |

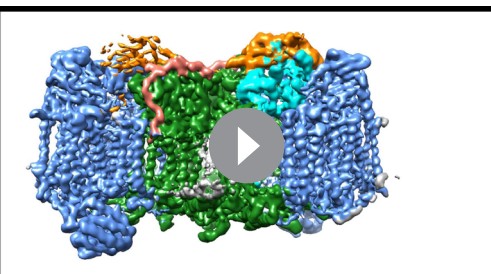

**Video 1.** Three-dimensional variability analysis of $CIII_2CIV_2$ showing the presence of LpqE with the cyt. *cc* subunit in the closed position or the absence of LpqE with the cyt. *cc* subunit in the open position. Subunits are colored as in Figure 1. Please view as a loop. https://elifesciences.org/articles/71959/figures#video1

map with a slight rotation relative to how it was fit previously (*Figure 1C*, *right*). 3DVA (*Figure 1C*, *Video 2*) shows that SOD moves between the center of the complex, where it was observed previously (*Gong et al., 2018*; *Wiseman et al., 2018*) to immediately above heme $c_I$. This proximity suggests that SOD may indeed transfer electrons abstracted from superoxide in the periplasm of *M. smegmatis* to CIV to contribute to the PMF (*Gong et al., 2018*; *Wiseman et al., 2018*), although this hypothesis requires further testing. The overall resolution of the map, which is somewhat better than in previous studies, allowed refinement of an atomic model for $CIII_2CIV_2$ including residues T82, E83, A123 to D131, and S183 from LpqE, and residues H57 to G78 from MSMEG_4693 (also known as CtaJ), which could not be modeled previously (*Figure 1D*, *Figure 1—figure supplement 3*, and *Table 1*). The model shown in *Figure 1D* illustrates one cyt. *cc* domain in the closed position with LpqE present (*Figure 1D*, *right side: cyan and orange*) and the other cyt. *cc* domain in the open position without LpqE (*Figure 1D*, *left side: cyan*), although other combinations could also be modeled.

## Nanomolar telacebec inhibits oxidoreductase activity with purified $CIII_2CIV_2$

To investigate inhibition of $CIII_2CIV_2$ by telacebec, we established a supercomplex activity assay, based on measurement of oxygen consumption with a Clark-type electrode. The mycobacterial electron transport chain uses $MQH_2$ as the electron donor for $CIII_2CIV_2$ while in canonical mitochondrial electron transport chains $UQH_2$ donates electrons to $CIII_2$ (*Cook et al., 2017*). Both $UQH_2$ and $MQH_2$ are insoluble in aqueous solution and consequently soluble analogues must be employed as substrates in assays with detergent-solubilized enzymes.

The midpoint potentials of the redox centers in mycobacterial $CIII_2$ are lower than those of canonical mitochondrial $CIII_2$ (*Kao et al., 2016*), and as a result $UQH_2$ analogues typically used in $CIII_2$ assays are not able to reduce $CIII_2$ of the *M. smegmatis* supercomplex. $MQH_2$ analogues (*Figure 2—figure supplement 1A-C*) capable of reducing $CIII_2CIV_2$ suffer from autoxidation at neutral or basic pH, which leads to oxygen reduction even in the absence of enzyme (*Munday, 2000*). This background oxygen-reduction rate is typically subtracted from oxygen reduction observed in the presence of enzyme to calculate the enzyme-catalyzed oxidoreductase activity. Previous measurement of $CIII_2CIV_2$ activity employed 2-methyl-[1,4]naphthohydroquinone (menadiol) (*Gong et al., 2018*) or 2,3-dimethyl-[1,4] naphthohydroquinone ($DMWH_2$) (*Graf et al., 2016*; *Wiseman et al., 2018*) as the electron donor. Although both substrates are susceptible to autoxidation, the rate of autoxidation was proposed to be ~30 % slower for $DMWH_2$ compared to menadiol at the pH 7.5 of our oxygen consumption assays (*Munday, 2000*). In contrast to these earlier studies, we found that menaquinol autoxidized more slowly than $DMWH_2$ (*Figure 2—figure supplement 1D*). However, we also found that mendiol was substantially less efficient than $DMWH_2$ as an electron donor for $CIII_2CIV_2$, likely due to its less favorable redox potential (*Fieser and Fieser, 1934*), supporting the choice of $DMWH_2$ as the substrate in assays (*Figure 2—figure supplement 1D*).

Initial activity assays led to anomalous results where addition of low concentrations of $CIII_2CIV_2$ to the assay mixture appeared to decrease the

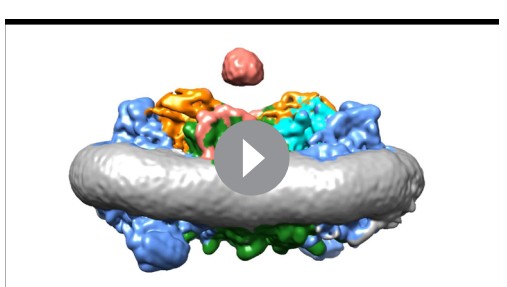

**Video 2.** Three-dimensional variability analysis showing movement of superoxide dismutase (SOD) subunit of $CIII_2CIV_2$. Subunits are colored as in Figure 1. Please view as a loop. https://elifesciences.org/articles/71959/figures#video2

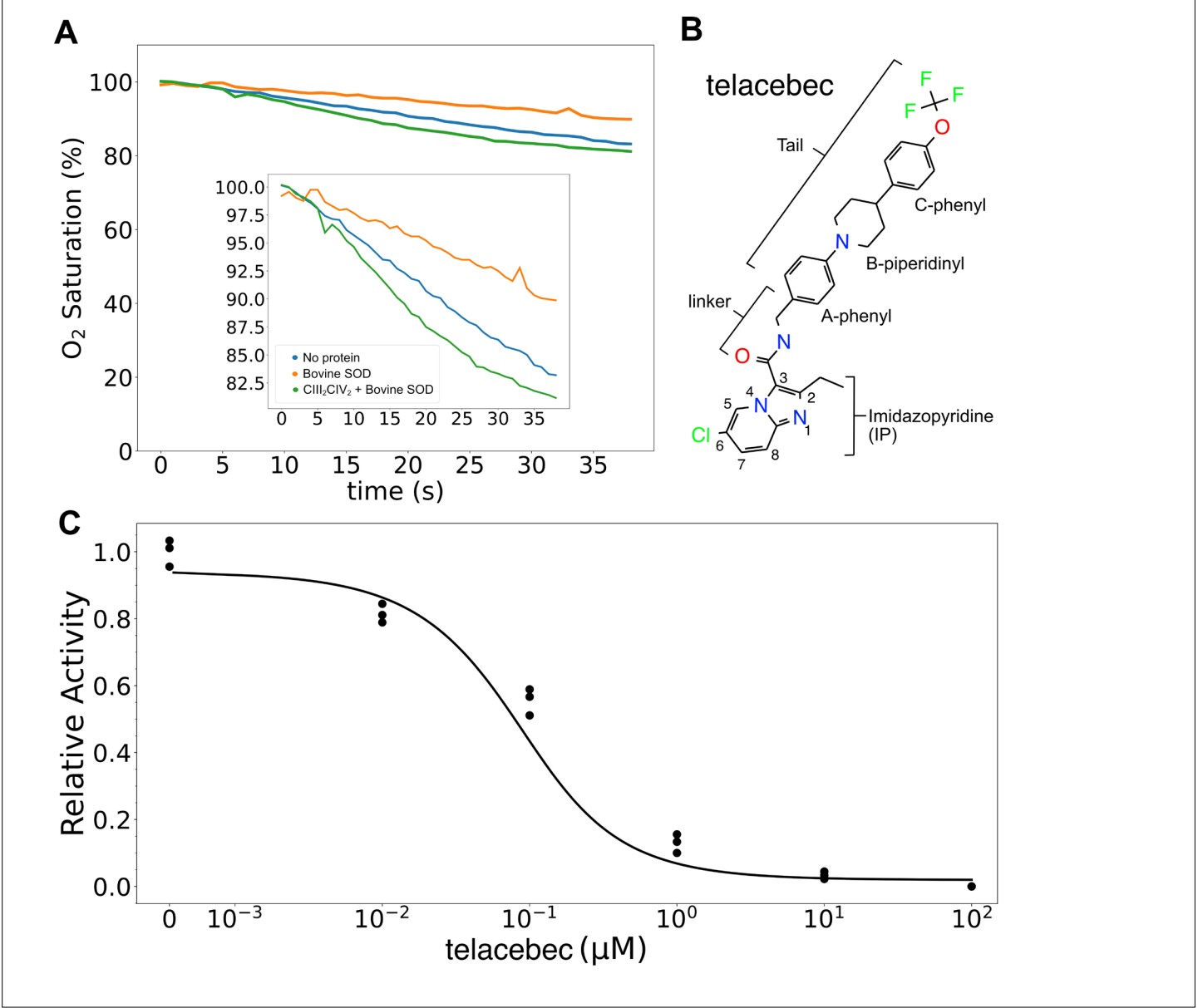

**Figure 2.** Assay of 2,3-dimethyl-[1,4]naphthohydroquinone (DMWH$_2$:O$_2$) oxidoreductase activity of CIII$_2$CIV$_2$. (**A**) An oxygen reduction assay shows that autoxidation of DMWH$_2$, *blue curve*, is decreased by the presence of 500 nM bovine superoxide dismutase (SOD), *orange curve*. Measurement of oxygen reduction by CIII$_2$CIV$_2$ in the presence of bovine SOD, *green curve*, allows calculation of CIII$_2$CIV$_2$ activity. (**B**) Structure of CIII$_2$CIV$_2$ inhibitor telacebec (Q203). (**C**) Titration of CIII$_2$CIV$_2$ (60 nM) with telacebec shows an IC$_{50}$ of 53 ± 19 nM (± s.d., n = 3 independent titrations) with 100 μM DMWH$_2$.

The online version of this article includes the following figure supplement(s) for figure 2:

**Figure supplement 1.** Structural and kinetic comparison of menaquinone and analogues.

rate of oxygen reduction below the background autoxidation rate. On subsequent investigation, we realized that at low concentrations of CIII$_2$CIV$_2$ the SOD subunit suppresses autoxidation of DMWH$_2$ more than CIII$_2$CIV$_2$ catalyzes oxidation of DMWH$_2$. This suppression of quinol autoxidation by SOD, which has been described previously (*Cadenas et al., 1988*), can lead to apparent negative activities for CIII$_2$CIV$_2$ when the background autoxidation is subtracted. Autoxidation of quinols is believed to involve a superoxide anion intermediate, with the SOD-catalyzed dismutation of the intermediate to hydrogen peroxide removing this reactant to slow the process (*Munday, 2000*). To remove error introduced by the effect of the SOD subunit on the observed oxygen reduction signal, we established that bovine C-type SOD can similarly limit the autoxidation of DMWH$_2$ (*Figure 2A*, *blue* and *orange*

curves), as well as menadiol (*Figure 2—figure supplement 1D*). Thus, by adding an excess of exogenous bovine SOD to assays, the $CIII_2CIV_2$'s $DMWH_2:O_2$ oxidoreductase activity can be measured with suppression of $DMWH_2$ autoxidation (e.g. *Figure 2A*, *green*). With 500 nM SOD added, the $CIII_2CIV_2$'s $DMWH_2:O_2$ oxidoreductase activity was measured at 91 ± 4 $e^-$/s (± s.d., n = 6 independent assays with three each from two separate batches of protein), which is nearly an order of magnitude greater than the apparent activity found previously (*Wiseman et al., 2018*). We subsequently added 500 nM bovine SOD to all assays to limit autoxidation of $DMWH_2$.

Telacebec (*Figure 2B*) is a potent inhibitor of mycobacterial $CIII_2$ (*Pethe et al., 2013*). The compound consists of an imidazo[1,2 -a]pyridine attached via an amide linker to an *N*-[(4-{4-[4-(trifluoromethoxy) phenyl]piperidin-1-yl}phenyl)methyl] 'tail'. Titrations of $CIII_2CIV_2$ activity with varying concentrations of telacebec (*Figure 2C*) show an $IC_{50}$ of 53 ±19 nM (± s.d., n = 3 independent titrations, with two titrations from one batch of purified protein and a third titration from a second batch of purified protein) with 65 nM $CIII_2CIV_2$ and 100 µM $DMWH_2$. This $IC_{50}$ is lower than the 840 ± 22 nM seen with the menadiol-based assay (*Gong et al., 2018*), but higher than the 20 nM concentration needed to inhibit 50 % of respiratory chain activity with inverted membrane vesicles from *M. smegmatis* (*Lu et al., 2018*) or 2.7 nM required to inhibit the 50 % of *M. tuberculosis* growth in liquid culture (*Pethe et al., 2013*). The increased $IC_{50}$ in the current assay compared to assays with inverted membrane vesicles or bacterial growth in liquid culture may be due to the binding affinity or high concentration of $DMWH_2$, which could allow $DMWH_2$ to compete with telacebec for binding to the complex. In addition, differences in inhibition in the different assays could be due to $CIII_2CIV_2$ being in detergent micelles rather than a lipid bilayer. The *M. tuberculosis* telacebec resistance mutations T313A and T313I (*Pethe et al., 2013*), equivalent to mutation of Thr308 in *M. smegmatis*, are near the $Q_P$ site and suggest that the inhibitor could interfere with $MQH_2$ binding to $CIII_2CIV_2$.

## The $CIII_2CIV_2$ structure has endogenous MQ in its $Q_P$ site

As telacebec is expected to bind near the $Q_P$ site of $CIII_2CIV_2$, we carefully characterized this site in the cryoEM map of the enzyme in the absence of inhibitor. The $Q_P$ site is near the periplasmic side of the membrane, located between heme $b_L$ and the FeS cluster (*Figure 3A*), and is formed by several loops and α helices from both the QcrB and QcrA subunits (*Figure 3B*). The arrangement of structural elements in the site is conserved from other CIIIs (*Sarewicz and Osyczka, 2015*). The entrance to the $Q_P$ site is formed by the C and F transmembrane α helices, and the cd1 α helix that separates the periplasmic side of the $Q_P$ pocket from the QcrA subunit. The ef helix and ef loop from the QcrB subunit are deeper in the $Q_P$ site, as is a short section from the QcrA subunit that includes the FeS-bound His368 residue (*Figure 3B*). His368 from QcrA is believed to have an important role in CIII, accepting a proton during quinol oxidation at the $Q_P$ site (*Mulkidjanian, 2005*).

In the inhibitor-free structure there is density for endogenous MQ in the $Q_P$ site (*Figure 3B*, *pale blue surface*). With the standard deviation of the cryoEM map normalized to σ = 1, the head group of MQ matches the density at 4.4σ. However, even with this strong density, the symmetry of the head group (*Figure 2—figure supplement 1*) makes it difficult to determine which of two poses, related by a 180° rotation, is correct. This ambiguity is exacerbated by weak density for the MQ tail, which is visible at 2.6σ, closer to the 1.7σ threshold used for visualizing lipids in the map. *Figure 3B* depicts the MQ pose that appears to match the density slightly better than the rotated pose, and is also the same pose as modeled previously (*Gong et al., 2018*). It is also possible that MQ could bind the structure in either pose, with the experimental map showing the average of both orientations.

The naphthoquinone head group of MQ is positioned near the entrance to the site, between the F and C helices (*Figure 3A*). This position for endogenous MQ was reported in a previous study of $CIII_2CIV_2$ from *M. smegmatis* (*Gong et al., 2018*). In this position, the naphthoquinone head group is ~14 Å away from the FeS cluster and the hydroxyl proton is ~15 Å from His368, which is too far for rapid coupled electron and proton transfer from $MQH_2$ to FeS and His368, respectively. This distance contrasts the deeper binding position adopted by UQ in ovine $CIII_2$ (*Letts et al., 2019*). It is also further from the FeS than the position observed for UQ-analogue inhibitors such as stigmatellin bound to chicken $CIII_2$ (*Zhang et al., 1998*), as well as 5-undecyl-6-hydroxy-4,7-dioxobenz othiazole (*Esser et al., 2004*) and 2-*n*-nonyl-4-hydroxyquinoline N-oxide (*Gao et al., 2003*) bound to bovine $CIII_2$. In the deeper position the head groups of UQ or its analogues are wedged between the ef helix/loop, and the cd1 helix, with the tail trailing between the F and C helix at the $Q_P$ site

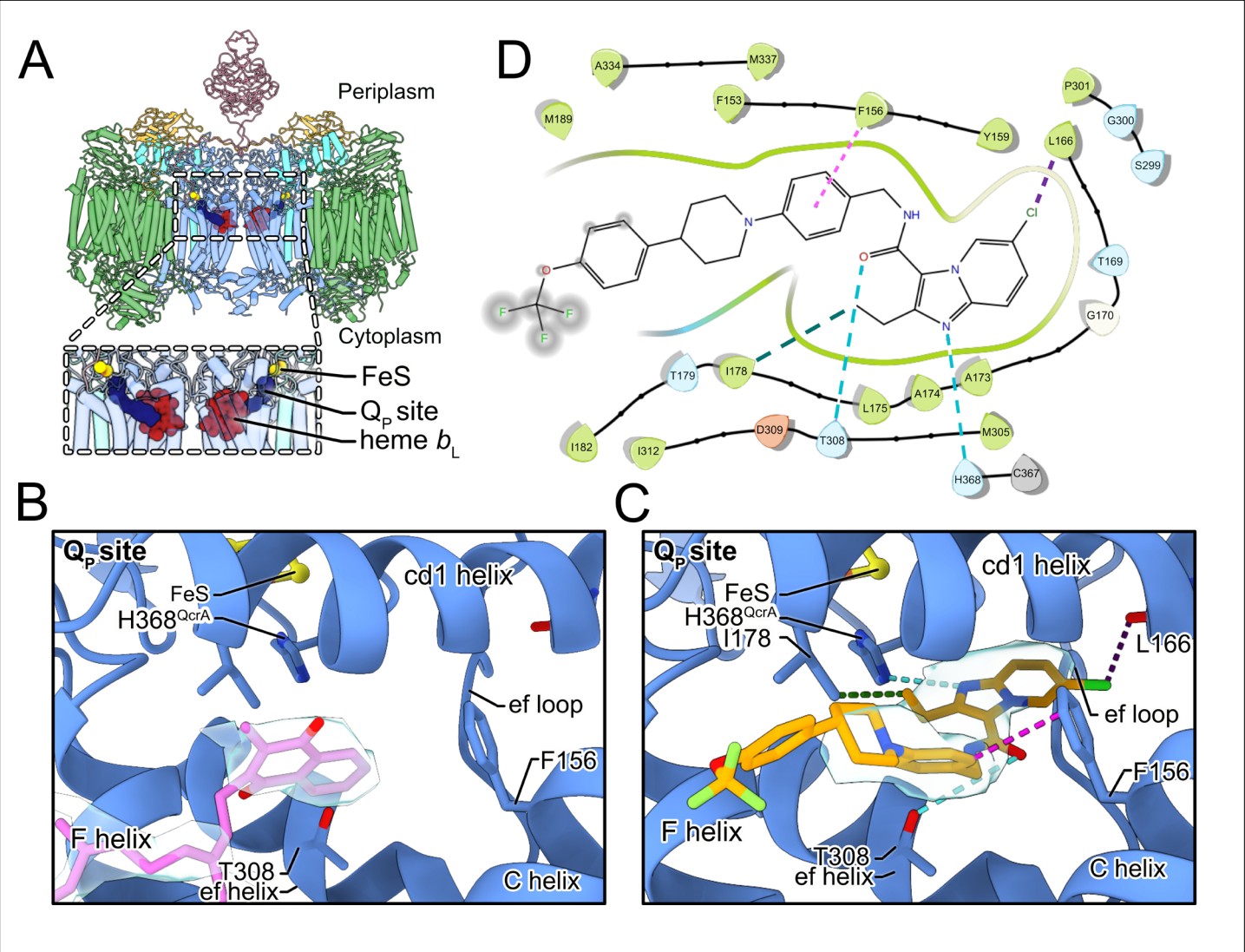

**Figure 3.** Telacebec binding to the $Q_P$ site. (**A**) The dashed boxes indicate the two $Q_P$ sites in $CIII_2CIV_2$, each showing menaquinone (*blue*), the Rieske protein FeS (*yellow*), and heme $b_L$ (*red*). (**B**) In the inhibitor-free structure, there is density for endogenous menaquinone (*pink model* and *gray surface*) distal from the FeS group among the well-conserved structural elements of the $Q_P$ site. (**C**) In the inhibitor-bound structure, there is density for telacebec (*orange model* and *gray surface*) deeper in the $Q_P$ site where it can form numerous interactions with the protein, including possible hydrogen bonds (*dashed teal lines*), hydrophobic interactions (*dashed green line*), a halogen bond (*dashed purple line*), and an aromatic interaction (*dashed pink line*). (**D**) A two-dimensional (2D) representation of the interactions between telacebec and residues of $CIII_2CIV_2$ using the same color convention as in part C.

The online version of this article includes the following figure supplement(s) for figure 3:

**Figure supplement 1.** Binding of small molecules in the $Q_P$ site of CIII.

entrance (*Figure 3—figure supplement 1A*). In these structures, the distance between the quinone head group and the FeS cluster depends on the position of the mobile Rieske head domain, but can be as little as ~7 Å, allowing for rapid electron transfer from the $UQH_2$ to the FeS (*Moser et al., 2006*).

### Telacebec replaces MQ in the $Q_P$ site of active $CIII_2CIV_2$

Our initial attempts to image $CIII_2CIV_2$ with telacebec failed to resolve the inhibitor (*Figure 3—figure supplement 1B*, *left*), leading us to consider the possibility that inhibitor binding occurs during substrate turnover by the enzyme. CryoEM of $CIII_2CIV_2$ in the presence of $DMWH_2$ but without telacebec confirmed that under these conditions the density in the $Q_P$ site was indistinguishable from MQ seen with the enzyme at rest (*Figure 3—figure supplement 1B*, *right*). We then incubated $CIII_2CIV_2$ with both $DMWH_2$ and telacebec to produce an inhibited complex and determined the structure of

this complex to a nominal resolution of 3.0 Å by cryoEM (*Figure 1—figure supplement 1*, *Table 1*). Telacebec binding did not cause large-scale conformational changes in $CIII_2CIV_2$ but produced a clear density for the inhibitor in each of the two $Q_P$ sites in the $CIII_2$ dimer (*Figure 3C*). The inhibitor's imidazopyridine moiety, amide linker region, and A-phenyl and B-piperidinyl moieties are all resolved clearly, with weaker density toward the end of the tail, which points into the lipid bilayer toward the cytoplasmic side of the membrane (*Figure 3C*, *pale blue surface*). These densities show that telacebec binds with its head group deep within the $Q_P$ binding pocket in a pose similar to UQ and the UQ-analogue inhibitors bound within the canonical $CIII_2$ as described above (*Figure 3—figure supplement 1A*). Telacebec's imidazopyridine moiety forms multiple interactions with the protein to stabilize inhibitor binding. Although hydrogen bonds cannot be detected with complete confidence at the present resolution, the position of the N1 nitrogen in telacebec's imidazopyridine moiety is consistent with formation of a hydrogen bond with the His368 from the QcrA subunit, which also binds the FeS group (*Figure 3C and D*, *dashed teal line*). The occurrence of a similar hydrogen bond between UQ and the equivalent histidine residue in canonical $CIII_2$ (*Zhang et al., 1998*) has been proposed to coordinate the Q cycle (e.g. see *Sarewicz and Osyczka, 2015*). The 2-ethyl group from the imidazopyridine is ~4 Å away from Ile178 from the QcrB subunit, providing hydrophobic interactions (*Figure 3C and D*, *dashed green line*), while the 6-chloro group from the imidazopyridine is close to the backbone carboxyl group of Leu166 from the QcrB subunit, enabling formation of a possible halogen bond (*Figure 3C and D*, *dashed purple line*).

In addition to its imidazopyridine moiety, the amide linker and tail of telacebec also contact subunit QcrB, stabilizing binding. Thr308, which is homologous with *M. tuberculosis* Thr313 and is known to be important for binding (*Pethe et al., 2013*), is <4 Å away from the linker region of telacebec. Although the rotameric state of Thr308 is ambiguous at the current resolution, one of the rotamer states could form a stabilizing hydrogen bond with the carbonyl group of the linker region (*Figure 3C and D*, *dashed teal line*). Finally, Phe156 is ~3.5 Å from the A-phenyl group of telacebec, allowing for aromatic-aromatic interaction between the protein and inhibitor (*Burley and a, 1985*; *Figure 3C and D*, *dashed pink line*) Interestingly, in the inhibitor-free specimen, the MQ head group is positioned similarly to the A-phenyl ring of telacebec and may form similar stabilizing contacts with the QcrB subunit (*Figure 3B and C*).

## Discussion

With telacebec and congeners having nanomolar inhibitory activity for both $CIII_2CIV_2$ and *M. tuberculosis* growth *in vitro*, antimycobacterial activity need not be improved for therapeutic purposes. However, the structural analysis reported here provides constraints and minimal requirements for activity of imidazopyridines and isosteric heterocycles with improved pharmacokinetic and physicochemical properties. Optimized physicochemical properties are important for drug production, including synthesis and purification. Improved pharmacokinetic properties could be enabled by design of analogues that retain target activity but are not recognized by mycobacterial efflux pumps, which are known to remove telacebec from bacterial cells to attenuate its antimycobacterial activity (*Jang et al., 2017*).

The structure also suggests how mutations can provide resistance to telacebec and why telacebec selectively inhibits mycobacterial $CIII_2$. The mutation T313A in *M. tuberculosis* (T308A in *M. smegmatis*) confers resistance to telacebec (*Pethe et al., 2013*), likely by removing the stabilizing hydrogen bond with the carbonyl group from the linker region of the inhibitor proposed above (*Figure 3C and D*). *M. smegmatis* grown in the presence of the telacebec analogue TB47 developed the mutation H190Y (*Lu et al., 2019*), which is adjacent to the cd1 helix and may alter the shape of the $Q_P$ binding site (*Figure 3—figure supplement 1C*). The selectivity of telacebec for mycobacterial $CIII_2CIV_2$ may derive, in part, from the lack of Thr308 in mammalian mitochondrial $CIII_2$ (*Figure 3—figure supplement 1D*). In addition, there may be clashes between the rigid telacebec tail and both Leu150 and Ile146 in mammalian $CIII_2$ (bovine numbering) due to the different location of the cd1 helix and the bulkier side chains in this region of the mammalian protein (*Figure 3—figure supplement 1D*). Interestingly, the mutation I147F (*M. smegmatis* numbering) results in resistance to the inhibitor stigmatellin in the *Saccharomyces cerevisiae* $CIII_2$ (*di Rago et al., 1989*).

The telacebec-bound structure also provides insight into the basic mechanism of $MQH_2:O_2$ oxidoreductase activity by $CIII_2CIV_2$. Multiple MQ binding sites were modeled in an earlier 3.5 Å resolution

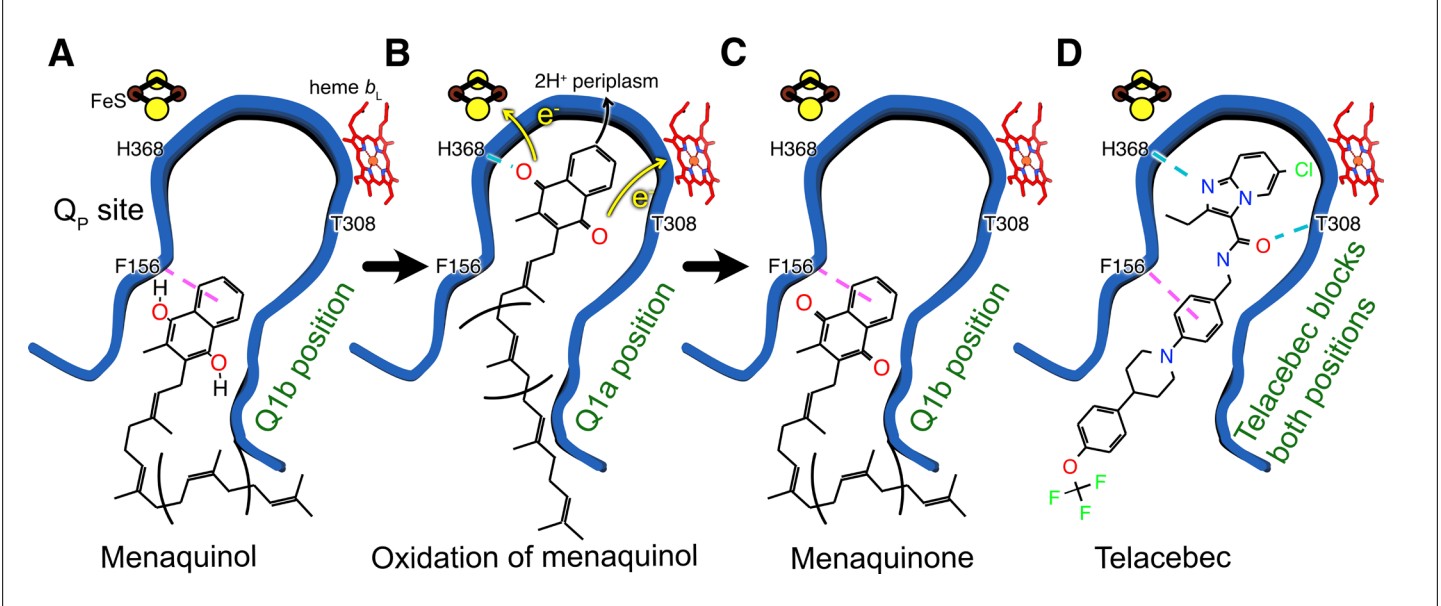

**Figure 4.** Model for oxidation of MQH$_2$ in the Q$_P$ site and how telacebec blocks it. An emerging model for MQH$_2$ reduction at the Q$_P$ site proposes that the substrate binds in the Q1b position where it is too far from FeS to donate protons and electrons (**A**). Upon moving deeper into the Q$_P$ site to the Q1a position, MQH$_2$ is oxidized to menaquinone (MQ), donating its first electron to FeS, its second electron to heme $b_L$, and releasing two protons to the positively charged periplasmic side of the lipid bilayer (**B**). Telacebec binds deep within the Q$_P$ site, forming numerous interactions with the protein and blocking both the Q1a and Q1b positions (**C**).

cryoEM density map of CIII$_2$CIV$_2$, but the significance of these sites was not clear (*Gong et al., 2018*). The structure presented here shows that telacebec, which can inhibit MQH$_2$:O$_2$ oxidoreductase activity completely (*Figure 2C*), binds only at the Q$_P$ site. Therefore, it is unlikely that MQ binding other than within the Q$_P$ site is involved in electron transfer to oxygen. During the Q cycle, two molecules of MQH$_2$ are oxidized to MQ at the Q$_P$ site for each molecule of MQ reduced to MQH$_2$ at the Q$_N$ site. It is possible that MQ is channeled between the Q$_P$ and Q$_N$ sites by staying loosely bound to the supercomplex surface, with the additional MQ sites serving as intermediate positions along the channeling pathway. A similar model was suggested for the spinach $b_6f$ complex, which is structurally and functionally related to CIII and carries out a Q cycle in plant chloroplasts with the hydrophobic electron carrier plastiquinone (*Malone et al., 2019*). It is also possible that these alternative MQ sites sequester MQ, increasing the local concentration of substrate near its two binding sites, which has been seen to increase the local concentration of ligands in other systems (*Vauquelin and Charlton, 2010*).

Within the Q$_P$ site different positions for both UQ and MQ have been described previously, both for canonical CIII and for CIII within a CIII$_2$CIV$_2$ supercomplex, respectively (*Ding et al., 1992*; *Moe et al., 2021*). Although the endogenous MQ found in the structure is most likely oxidized, the position it occupies is along the access path to the FeS center. We speculate that reduced MQH$_2$ also adopts the same position, at least transiently, during turnover. This binding site within Q$_P$ is denoted the Q1b position. As discussed above, the Q1b position is too far for rapid transfer of protons and electrons from MQH$_2$ to His368 and the FeS center, respectively (*Figure 4A*). In contrast, movement of MQH$_2$ deeper within the Q$_P$ site to a Q1a position would bring it less than 10 Å from FeS, close enough to donate electrons to the redox center and protons to His368 (*Figure 4B*). The oxidized MQ could then return to the Q1b position, as observed in the present structure (*Figure 4C*), before it is exchanged for MQH$_2$ from the membrane. In an emerging model for CIII$_2$CIV$_2$ function, the Q1b position serves as a 'stand-by' site for MQH$_2$, with oxidation of the substrate occurring only upon relocation to Q1a (see *Moe et al., 2021*; *Mulkidjanian, 2005*). The structure of CIII$_2$CIV$_2$ with telacebec bound shows that the compound serves as dual site inhibitor (*Figure 4D*), with the imidazopyridine group bound to the Q1a position and A-phenyl portion of the tail bound to the Q1b position. Indeed, the possible hydrogen bond between Thr308 and the carboxyl group in the linker region of telacebec could also

occur between Thr308 and MQ, stabilizing it in the Q1b position. Therefore, the observed pose of the inhibitor within the enzyme not only blocks MQ access to the FeS center but fills the $Q_P$ site entirely.

Inhibiting $CIII_2CIV_2$ has demonstrated antituberculosis activity in humans (*de Jager et al., 2020*), even though *M. tuberculosis* possesses cyt. *bd* as an alternative enzyme that can oxidize $MQH_2$ and sustain the electron flux to oxygen in the mycobacterial membrane. More potent killing of mycobacterial pathogens can be accomplished by simultaneous inhibition of both the $CIII_2CIV_2$ and cyt. *bd* terminal oxidases (*Beites et al., 2019*; *Kalia et al., 2017*; *Lee et al., 2021*). The present study demonstrates how cryoEM can reveal the mechanisms of electron transport chain inhibitors, enabling new strategies for targeting mycobacterial infections.

## Materials and methods

### Construction of *M. smegmatis* strain, cell culture, and protein isolation

An *M. smegmatis* strain with a 3×FLAG tag at the C terminus of subunit QcrB was generated with the ORBIT method (*Murphy et al., 2018*). This method requires transformation of the parent strain with a plasmid encoding the Che9c phage RecT annealase and Bxb1 integrase, a payload plasmid with the desired insert, and an oligonucleotide that guides integration of the payload into the chromosomal DNA. The parent strain MC2155 was transformed with plasmid pKM444, which encodes the Che9c annealase and Bxb1 integrase. The resulting strain was subsequently transformed with payload plasmid pSAB41, which encodes a 3×FLAG tag and was described previously (*Guo et al., 2021*), as well as the targeting oligonucleotide 5′-CAAGTCGCTCACGGCGCTCAAGGAGCACCAGGACCG CATCCACGGCAACGGGGAGACCAACGGTCATCACGGTTTGTCTGGTCAACCACCGCGGTCTCAG TGGTGTACGGTACAAACCTGATCGCTGAGATACTCGGATCGCCGCAATTCCTCTTCGGAGGGGT TGCGGCGATCTTTTTATGTGCGCTTTC-3′. The resulting strain '*M. smegmatis* QcrB-3xFLAG' was selected with hygromycin (50 μg/mL) and correct insertion of the 3×FLAG sequence was confirmed by colony PCR.

*M. smegmatis* was cultured in 7H9 medium (Sigma) supplemented with TDS (10 g/L tryptone, 2 g/L dextrose, 0.8 g/L NaCl). A preculture in liquid medium (15 mL) was inoculated with cells from an agar plate and grown at 30 °C with shaking at 180 rpm for 48 hr. This culture was used to inoculate a larger culture (6 L), which was grown at the same conditions for a further 48 hr. Cells were harvested by centrifugation at 6900× *g* for 20 min and frozen in liquid nitrogen for subsequent use. After thawing, cells were broken by four passes through a continuous flow cell disruptor (Avestin) at 20 kpsi and membranes were harvested by centrifugation at 39,000× *g* for 30 min.

To purify $CIII_2CIV_2$, membranes were resuspended in lysis buffer (50 mM Tris-HCl pH 7.5, 100 mM NaCl, 0.5 mM EDTA) at 4 mL/g and solid dodecyl maltoside (DDM) detergent was added to 1 % (w/v) with stirring at 4 °C for 45 min. Insoluble material was removed by centrifugation at 149,000× *g* for 45 min and the solubilized protein was loaded onto a gravity column of 2 mL M2 anti-FLAG affinity matrix (Sigma). The column was washed with 10 mL of wash buffer (50 mM Tris-HCl pH 7.4, 150 mM NaCl, 0.02 % [w/v] DDM) and eluted with 5 mL of wash buffer supplemented with 3×FLAG peptide at 150 μg/mL. Purified protein was exchanged into 50 mM Tris-HCl pH 7.4, 150 mM NaCl, and 0.003% (w/v) glycol-diosgenin (GDN) with a 100 kDa molecular weight cutoff concentrator (Sigma).

### Activity assays

DMW (Enamine) at 20 mM in anhydrous ethanol (400 μL) on ice was reduced with a few grains of $NaBH_4$ and the reaction was quenched by addition of 4 N HCl (10–20 μL). Enzymatically reduced $DMWH_2$ was prepared with 7.5 mM NADH, 200 DMW, and 300 μg/mL NDH-2 from *Caldalkalibacillus thermarum* (*Nakatani et al., 2017*). Oxygen-reduction assays were performed with an Oxygraph Clark-type electrode (Hansatech) in 1 mL of reaction buffer (50 mM Tris-HCl pH 7.5, 100 mM NaCl, 0.5 mM EDTA, and 500 nM bovine SOD [Sigma]). $CIII_2CIV_2$ was added (65 nM) and reactions were initiated by addition of 100 μM $DMWH_2$. For inhibition studies, telacebec (DC Chemicals) at varying concentrations was incubated with 65 nM $CIII_2CIV_2$ in the reaction buffer for 3 hr at 4 °C. This mixture was added to the Oxygraph and reactions were initiated by addition of 100 μM $DMWH_2$. To account for any background oxygen reduction that still occurs in the presence of SOD, the rate of oxygen reduction in the presence of $DMWH_2$ and SOD, but in the absence of $CIII_2CIV_2$, was subtracted from the rate in the presence of $DMWH_2$, SOD, and $CIII_2CIV_2$. The resulting oxygen reduction rates for

$CIII_2CIV_2$ at different concentrations of telacebec were fit with a Python script. Individual inhibition curves, which were produced on different days with different preparations of reagent, were fit individually, with the average of the $IC_{50}$ values reported and the standard deviation of the fitted $IC_{50}$ values reported as the error (*Dahlin et al., 2004*). Plots were produced using the Python matplotlib library.

## CryoEM specimen preparation and imaging

For cryoEM of inhibitor-free $CIII_2CIV_2$, enzyme at ~16 mg/mL (2 μL) was applied to homemade nano-fabricated holey gold grids (*Marr et al., 2014*), which had previously been glow-discharged in air for 120 s at 20 mA (PELCO easiGlow), within a Vitrobot Mark III (FEI) at 4 °C and 100 % relative humidity. Grids were blotted for 24 s before freezing. For cryoEM of telacebec-bound $CIII_2CIV_2$, $DMWH_2$ in ethanol was added to 100 μM (0.02 % ethanol) and telacebec in DMSO was added to 25 μM (1.5 % DMSO) to a solution containing purified $CIII_2CIV_2$ at ~0.08 mg/mL (6 mL). The solution was concentrated ~100 -fold by centrifugation at 700× *g* with a 100 kDa molecular weight cutoff centrifuged concentrator device (Sigma). The sample (2 μL) was then applied to homemade nanofabricated holey gold grids, which had previously been glow-discharged in air for 120 s at 20 mA, within an EM GP2 (Leica) grid freezing device at 4 °C and 100 % relative humidity. Grids were blotted for 1 s before freezing.

Screening of specimens was done with an FEI Tecnai F20 electron microscope equipped with a K2 Summit direct detector device camera. High-resolution cryoEM data were collected with a Titan Krios G3 electron microscope (Thermo Fisher Scientific) operated at 300 kV and equipped with a Falcon 4 direct detector device camera. Automated data collection was done with the EPU software package. The inhibitor-free dataset consisted of 2793 movies and telacebec-bound sample consisted of 4308 movies. Movies were collected at a nominal magnification of 75,000× with a calibrated pixel size of 1.03 Å. Movies consisted of 30 exposure fractions over 7.7 s. The camera exposure rate and the total exposure were 5.99 e⁻/pixel/s and ~43.5 e⁻/Å², respectively (*Table 1*).

## Image analysis and atomic model building

All image analysis was performed within the *cryoSPARC* software package, version 3 (*Punjani et al., 2017*), including individual particle motion correction (*Rubinstein and Brubaker, 2015*), non-uniform refinement (*Punjani et al., 2020*), and 3DVA (*Punjani and Fleet, 2021*). Image analysis and 3D reconstruction for each dataset was performed in the same way. Motion was corrected and CTF parameters were estimated for each movie in patches. Manual particle selection and 2D classification was used to generate templates, which were in turn used to select of 387,777 and 1,037,709 particle images for the telacebec-bound and inhibitor-free datasets, respectively. Datasets were cleaned with 2D classification, 3D classification, and heterogeneous refinement to 70,818 and 150,885 particle images for the telacebec-bound and inhibitor-free datasets, respectively. Beam tilt was corrected and each map was refined with non-uniform refinement without symmetry enforced. CTF values were then refined, the detergent micelle subtracted, and alignment parameters adjusted with local refinement with C2 symmetry enforced, yielding maps at 3.0 Å resolution for each dataset. 3DVA was done on a pooled dataset with masks including the SOD subunit or cyt. *cc* domain. Atomic models were constructed starting from previous models of the complex (*Gong et al., 2018*; *Wiseman et al., 2018*). Additions to the models were made in *Coot* (*Emsley et al., 2010*) and refined with *Phenix* (*Liebschner et al., 2019*) and *ISOLDE* (*Croll, 2018*).

## Structure-activity relation analysis studies

Insight into protein-inhibitor interaction was facilitated by analysis with the *Schrödinger* software package (Release 2019–1). The protein preparation wizard within *Schrödinger* was used to prepare the protein for modeling. Briefly, the QcrA and QcrB chains and the ligand from the PDB file were merged and pre-processed to add missing hydrogen atoms, fill in missing side chains, and adjust ionization and tautomeric states of the ligand. The hydrogen bond network between the protein's amino acids and the ligand was optimized by allowing reorientation of amino acid side chains like His, Asn, Asp, Glu, and Gln, and the ionization and tautomeric states of these side chains were estimated (*Olsson et al., 2011*). The resulting structure was refined to remove clashes and optimize geometry with the OPLS3e force field (*Roos et al., 2019*). These changes did not noticeably affect the fit of the model within the experimental cryoEM density map.

## Acknowledgements

DJY was supported by a Canada Graduate Scholarship from the Canadian Institutes of Health Research (CIHR) and a Queen Elizabeth II Graduate Scholarship in Science and Technology from the University of Toronto Department of Medical Biophysics, JMDT was supported by a postdoctoral fellowship from the Canadian Institutes of Health Research, and JLR was supported by the Canada Research Chairs program. This research was supported by CIHR grant PJT162186 (JLR), The Alice and Knut Wallenberg Foundation grant 2019.0043 (PB), and Swedish Research Council grant 2018–04619 (PB). CryoEM data were collected at the Toronto High-Resolution High-Throughput Cryo-EM facility, supported by the Canada Foundation for Innovation and Ontario Research Fund.

## Additional information

### Funding

| Funder | Grant reference number | Author |
|---|---|---|
| Canadian Institutes of Health Research | PJT162186 | John L Rubinstein |
| Wallenberg Foundation | 2019.0043 | Peter Brzezinski |
| Swedish Research Council | 2018-04619 | Peter Brzezinski |
| Canadian Institutes of Health Research | PGS-M | David J Yanofsy |
| Canadian Institutes of Health Research | PDF | Justin M Di Trani |
| Canada Research Chairs | | John L Rubinstein |
| Canada Foundation for Innovation | | John L Rubinstein |
| Ontario Research Foundation | | John L Rubinstein |
| University of Toronto | | David J Yanofsy |

The funders had no role in study design, data collection and interpretation, or the decision to submit the work for publication.

### Author contributions

David J Yanofsky, Justin M Di Trani, Formal analysis, Investigation, Methodology, Visualization, Writing – original draft, Writing – review and editing; Sylwia Król, Formal analysis, Investigation, Methodology; Rana Abdelaziz, Formal analysis, Investigation, Methodology, Visualization; Stephanie A Bueler, Investigation, Methodology; Peter Imming, Formal analysis, Funding acquisition, Methodology, Supervision, Writing – review and editing; Peter Brzezinski, Conceptualization, Formal analysis, Funding acquisition, Methodology, Supervision, Writing – review and editing; John L Rubinstein, Conceptualization, Formal analysis, Funding acquisition, Methodology, Supervision, Writing – original draft, Writing – review and editing

### Author ORCIDs

Justin M Di Trani ⓘ http://orcid.org/0000-0003-4764-4009
Rana Abdelaziz ⓘ http://orcid.org/0000-0002-9581-604X
Peter Imming ⓘ http://orcid.org/0000-0003-2178-3887
Peter Brzezinski ⓘ http://orcid.org/0000-0003-3860-4988
John L Rubinstein ⓘ http://orcid.org/0000-0003-0566-2209

### Decision letter and Author response

Decision letter https://doi.org/10.7554/eLife.71959.sa1
Author response https://doi.org/10.7554/eLife.71959.sa2

## Additional files

### Supplementary files
• Transparent reporting form

### Data availability
Data deposition: all electron cryomicroscopy maps described in this article have been deposited in the Electron Microscopy Data Bank (EMDB) (accession numbers EMD-24455 to EMD-24457) and atomic models have been deposited in the Protein Database (PDB) (accession numbers 7RH5 to 7RH7).

The following dataset was generated:

| Author(s) | Year | Dataset title | Dataset URL | Database and Identifier |
|---|---|---|---|---|
| Di Trani JM, Yanofsky DJ, Rubinstein JL | 2021 | Mycobacterial CIII2CIV2 supercomplex, Inhibitor free | https://www.ebi.ac.uk/emdb/EMD-24455 | Electron Microscopy Data Bank, EMD-24455 |
| Di Trani JM, Yanofsky DJ, Rubinstein JL | 2021 | Mycobacterial CIII2CIV2 supercomplex, inhibitor free, -Lpqe cyt cc open | https://www.ebi.ac.uk/emdb/EMD-24456 | Electron Microscopy Data Bank, EMD-24456 |
| Di Trani JM, Yanofsky DJ, Rubinstein JL | 2021 | Mycobacterial CIII2CIV2 supercomplex, Telacebec (Q203) bound | https://www.ebi.ac.uk/emdb/EMD-24457 | Electron Microscopy Data Bank, EMD-24457 |
| Di Trani JM, Yanofsky DJ, Rubinstein JL | 2021 | Mycobacterial CIII2CIV2 supercomplex, Inhibitor free | https://www.rcsb.org/structure/7RH5 | RCSB Protein Data Bank, 7RH5 |
| Di Trani JM, Yanofsky DJ, Rubinstein JL | 2021 | Mycobacterial CIII2CIV2 supercomplex, inhibitor free, -Lpqe cyt cc open | https://www.rcsb.org/structure/7RH6 | RCSB Protein Data Bank, 7RH6 |
| Di Trani JM, Yanofsky DJ, Rubinstein JL | 2021 | Mycobacterial CIII2CIV2 supercomplex, Telacebec (Q203) bound | https://www.rcsb.org/structure/7RH7 | RCSB Protein Data Bank, 7RH7 |

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
