## [Decision Letter]

**Acceptance summary:**

The paper by Rubinstein and colleagues is the first study to elucidate the atomic-resolution structure of the major respiratory complex of *Mycobacterium tuberculosis* inhibited by Telacebec (Q203), a novel first-in-class proven antituberculosis drug currently in phase II clinical trials. The work provides a new molecular blue print for improving and developing more inhibitors of this essential respiratory complex in the fight against the global pandemic of drug resistant tuberculosis disease.

**Decision letter after peer review:**

Thank you for submitting your article "Structure of mycobacterial CIII_2_CIV_2_ respiratory supercomplex bound to the tuberculosis drug candidate telacebec (Q203)" for consideration by *eLife*. Your article has been reviewed by 3 peer reviewers, and the evaluation has been overseen by a Reviewing Editor and Olga Boudker as the Senior Editor. The following individual involved in review of your submission has agreed to reveal their identity: James A Letts (Reviewer #1).

Essential revisions:

1) Please correct geometry violations in the some of your lipids – see Reviewer 1 for details.

2) Ensure activity assays are performed with biological replicates (separate protein preparation etc). – see Reviewer 2 for details.

*Reviewer #1 (Recommendations for the authors):*

The discussion of the new activity assay conditions could be strengthened by a Figure supplement comparing the structures of menaquinone, menadiol and 2,3-dimethyl-[1,4]naphthpquinone. The relative importance of the switch from menadiol to 2,3-dimethyl-[1,4]naphthpquinone vs. the addition of SOD to the reaction conditions is unclear. Does the addition of SOD to the menadiol assay result in a similar reduction of autoxidation? If known, the proposed mechanism by which SOD slows autoxidation should also be briefly mentioned as this activity is key to the improved assay and interpretation of the results.

Videos from the 3D variability analysis showing the movement of the SOD subunit and loss of the LpqE subunit would aid this section.

Page 6, paragraph 2 the authors state "The increased IC50 in the current assay compared to assays with inverted membrane vesicles or bacterial growth in liquid culture may be due to the binding affinity or high concentration of DMW, which could allow DMW to compete with telacebec for binding to the complex." Are the authors able to use their assay to determine the KM for DMW? Have they attempted IC50 measurements at different DMW concentrations? Is the mode of telacebec inhibition well characterized? How would the differences between detergent and membrane environments impact the measured IC50?

On page 6, paragraph 4, sentence 1, the authors reference "strong density for the endogenous MQ", the term "strong density" is ambiguous and should be quantified. If the map is normalized, at what σ value is the density clearly visible? How does this σ value compare to those of other map features, such as heavy metals, core side chains, other lipids or the detergent micelle? When looking at the map it is difficult to trace many of the modeled lipid and isoprenoid tails in the density. Also the orientation of the menaquinone head group in the QP site appears ambiguous, the isoprenoid tail could be as modeled or could go towards Met189 with the menaquinone head flipped 180 degrees. Why did the authors choose the current orientation? Given the symmetry of the menaquinone head are both binding orientations possibly present and averaged out in the structure? Would this help explain the ambiguity in the density for the isoprenoid chain?

There are many geometry violations in the isoprenoid chains of the modeled menaquinones (see chain F MQ9 607 C34 for an extreme example), the authors should double check their ligand restraints file. Overall the density for chain F MQ9 607 is unconvincing that it belongs to a menaquinone. The density for the isoprenoid tail for chain F MQ9 609 is very weak and difficult to follow why it was traced in the way it was (regions of it are not in density, whereas significant density in the same vicinity remains unmodeled). This issue is also seen in other regions of the map/model, i.e. there are several clear densities for lipids that are not modeled and modeled acyl chains that are not in clear density. The authors should carefully reevaluate their lipid and menaquinone modeling before publication.

The authors first mention His368 on page 6 but do not discuss its significance until pages 7 and 8. Clarity could be improved by discussing the significance of His368 when it is first mentioned.

Given that the co-purified menaquinone seen in the inhibitor-free enzyme is likely oxidized, wouldn't the Q1b position more likely be for oxidized menaquinone as opposed to the reduced menaquinol as depicted in Figure 4A? If so wouldn't the menaquinone in the Q1b position likely be more representative "exiting" the QP-site after donating its electrons, as opposed to a menaquinol "entering" the QP-site in order to donate its electrons? If so, what would the implications be for the reversibility of the Q-cycle and semiquinone formation at the QP-site, to have a menaquinone binding site adjacent to the QP site that is too distance to donate electrons to or accept electrons from the FeS cluster or heme bL?

*Reviewer #2 (Recommendations for the authors):*

My only recommendation is using multiple biological replicates (at least two, better three) for the activity assays.

*Reviewer #3 (Recommendations for the authors):*

1. Readers will be interested to see how reported mutations in qcrB in *M. tuberculosis* that confer resistance to Q203 map onto the CIII2CIV2 inhibited complex. How does the H190Y mutation reported for TB47 map to the new structural data with the Q203-inhibited structure (see Lu et al., ACS Infectious Diseases 5:239-249 (2019) PMID: 30485737).

2. A short paragraph might be appropriate on how the new structural data reported herein could be used to improve on inhibitors of this supercomplex.

3. A structural evaluation of why Q203 is unable to inhibit the mitochondrial Complex III structure could be included.

4. The Beites et al., 2019 reference should also include these two citations:

Lee BS, et al., EMBO Molecular Medicine 13: e13207 (2021)

Kalia, NP, et al., Proceedings of the National Academy of Sciences of the United States of America 114:7426-7431 (2017)

---

## [Author Response]

Essential revisions:1) Please correct geometry violations in the some of your lipids – see Reviewer 1 for details.2) Ensure activity assays are performed with biological replicates (separate protein preparation etc). – see Reviewer 2 for details.Reviewer #1 (Recommendations for the authors):The discussion of the new activity assay conditions could be strengthened by a Figure supplement comparing the structures of menaquinone, menadiol and 2,3-dimethyl-[1,4]naphthpquinone. The relative importance of the switch from menadiol to 2,3-dimethyl-[1,4]naphthpquinone vs. the addition of SOD to the reaction conditions is unclear. Does the addition of SOD to the menadiol assay result in a similar reduction of autoxidation?

We thank the reviewer for this suggestion. Three figure supplement panels have been added (Figure 2 —figure supplement 1A-C) to show the structural differences between the different CIII_2_CIV_2_ substrates.

We had initially selected DMWH_2_ over menadiol based on DMWH_2_ having a more favorable redox potential and literature that states that menaquinol autoxidizes faster than DMWH_2_. The reviewer’s comment prompted us to compare menaquinol and DMWH_2_ experimentally and investigate the effect of SOD on autoxidation of menaquinol. We found that in our conditions, menaquinol autoxidizes more slowly than DMWH_2_ and the autoxidation can be limited by SOD, but even more importantly we confirmed in multiple experiments that menaquinol is not as effective as an electron donor for CIII_2_CIV_2_ as DMWH_2_. We have updated the relevant text (p. 5, 3^rd^ paragraph):

“The midpoint potentials of the redox centers in mycobacterial CIII_2_ are lower than those of canonical mitochondrial CIII_2_ (Kao et al., 2016), and as a result UQH_2_ analogues typically used in CIII_2_ assays are not able to reduce CIII_2_ of the *M.* *smegmatis* supercomplex. MQH_2_ analogues (Figure 2 —figure supplement 1A-C) capable of reducing CIII_2_CIV_2_ suffer from autoxidation at neutral or basic pH, which leads to oxygen reduction even in the absence of enzyme (Munday, 2000). This background oxygen-reduction rate is typically subtracted from oxygen reduction observed in the presence of enzyme to calculate the enzyme-catalyzed oxidoreductase activity. Previous measurement of CIII_2_CIV_2_ activity employed 2-methyl-[1,4]naphthohydroquinone (menadiol) (Gong et al., 2018) or 2,3-dimethyl-[1,4]naphthohydroquinone (DMWH_2_) (Graf et al., 2016; Wiseman et al., 2018) as the electron donor. Although both substrates are susceptible to autoxidation, the rate of autoxidation was proposed to be ~30% slower for DMWH_2_ compared to menadiol at the pH 7.5 of our oxygen consumption assays (Munday, 2000). In contrast to these earlier studies, we found that menaquinol autoxidized more slowly than DMWH_2_ (Figure 2 —figure supplement 1D). However, we also found that mendiol was substantially less efficient than DMWH_2_ as an electron donor for CIII_2_CIV_2_, likely due its less favorable redox potential (Fieser and Fieser, 1934), supporting the choice of DMWH_2_ as the substrate in assays (Figure 2 —figure supplement 1D).”

We have added a panel to Figure 2 – Supplement 1 that compares the behaviour of menadiol and DMWH_2_.

As seen in Figure 2 – Supplement 1, SOD does limit autoxidation of menaquinol (but does not solve the problem of menaquinol being a poor electron donor to CIII_2_CIV_2_). To make this point clear we have added the following text (p.5, 4^th^ paragraph):

“To remove error introduced by the effect of the SOD subunit on the observed oxygen reduction signal, we established that bovine C-type SOD can similarly limit the autoxidation of DMWH_2_ (Figure 2A, *blue* and *orange* curves), as well as menadiol (Figure 2 —figure supplement 1D).”

If known, the proposed mechanism by which SOD slows autoxidation should also be briefly mentioned as this activity is key to the improved assay and interpretation of the results.

The autoxidation of quinols is complex process, thought to involve as many as seven different steps in the reaction. A model for how SOD slows autoxidation has been proposed and we now provide a concise description of the process, along with a citation to the reference where it is described more fully (p. 5, 4^th^ paragraph):

“Autoxidation of quinols is believed to involve a superoxide anion intermediate, with the SOD-catalyzed dismutation of the intermediate to hydrogen peroxide removing this reactant to slow the process (Munday, 2000).”

Videos from the 3D variability analysis showing the movement of the SOD subunit and loss of the LpqE subunit would aid this section.

We agree with the reviewer and have now added the following supplementary videos and referred to them in the text where 3D variability is described:

Video 1. Three-dimensional variability analysis of CIII_2_CIV_2_ showing the presence of LpqE with the cyt. *cc* subunit in the closed position or the absence of LpqE with the cyt. *cc* subunit in the open position. Subunits are coloured as in Figure 1. Please view as loop.

Video 2. Three-dimensional variability analysis showing movement of SOD subunit of CIII_2_CIV_2_. Subunits are coloured as in Figure 1. Please view as loop.

Page 6, paragraph 2 the authors state "The increased IC50 in the current assay compared to assays with inverted membrane vesicles or bacterial growth in liquid culture may be due to the binding affinity or high concentration of DMW, which could allow DMW to compete with telacebec for binding to the complex." Are the authors able to use their assay to determine the KM for DMW? Have they attempted IC50 measurements at different DMW concentrations? Is the mode of telacebec inhibition well characterized? How would the differences between detergent and membrane environments impact the measured IC50?

Unfortunately, we are not able to reliably measure the KM for DMW due to technical limitations with the oxygen consumption assay. First, the assays are extremely low throughput, with each measurement done separately and requiring extensive measurement time and a large quantity of purified protein. Further, measuring the activity at lower DMW concentrations is both technically and practically prohibitivly challenging. Even though the autoxidation rate is decreased in the presence of SOD, it is not eliminated completely (see Figure 2A and Figure 2 —figure supplement 1). Lowering the DMW concentrations decreases the measured CIII_2_CIV_2_ activity such that it approaches the autoxidation rate, which introduces large errors into the determination of the activity. Furthermore, because each activity measurement requires monitoring the O_2_-reduction rate for tens of seconds, the simultaneous formation of oxidized DMW by autoxidation results in accumulation of oxidized DMW that, at low DMW concentrations, yields non-linear O_2_-oxidation kinetics and prevents reliable determination of the slope of the O_2_ concentration curve. While we agree that a highly detailed analysis of telacebec inhibition of CIII_2_CIV_2_ at varying conditions would be desirable, we consider measuring a reliable IC_50_ at even one condition an important advance.

The reviewer raises a good point that differences in IC_50_ can also be due to measuring activity in detergent micelles rather than lipid bilayers. Therefore, we have added the following text to introduce this possibility (p. 6, 2^rd^ paragraph):

“In addition, differences in inhibition in the different assays could be due to CIII_2_CIV_2_ being in detergent micelles rather than a lipid bilayer.”

On page 6, paragraph 4, sentence 1, the authors reference "strong density for the endogenous MQ", the term "strong density" is ambiguous and should be quantified. If the map is normalized, at what σ value is the density clearly visible? How does this σ value compare to those of other map features, such as heavy metals, core side chains, other lipids or the detergent micelle? When looking at the map it is difficult to trace many of the modeled lipid and isoprenoid tails in the density. Also the orientation of the menaquinone head group in the QP site appears ambiguous, the isoprenoid tail could be as modeled or could go towards Met189 with the menaquinone head flipped 180 degrees. Why did the authors choose the current orientation? Given the symmetry of the menaquinone head are both binding orientations possibly present and averaged out in the structure? Would this help explain the ambiguity in the density for the isoprenoid chain?

We thank the reviewer for these suggestions. We have modified the text as follows to address them (p.7, 1^st^ paragraph):

“In the inhibitor-free structure there is density for endogenous MQ in the Q_P_ site (Figure 3B, *pale blue surface*). With the standard deviation of the cryoEM map normalized to σ=1, the head group of MQ matches the density at 4.4σ. However, even with this strong density, the symmetry of the head group (Figure 2 —figure supplement 1) makes it difficult to determine which of two poses, related by a 180º rotation, is correct. This ambiguity is exacerbated by weak density for the MQ tail, which is visible at 2.6σ, closer to the 1.7σ threshold used for visualizing lipids in the map. Figure 3B depicts the MQ pose that appears to match the density slightly better than the rotated pose, and is also the same pose as modelled previously (Gong et al., 2018). It is also possible that MQ could bind the structure in either pose, with the experimental map showing the average of both orientations.”

This paragraph makes clear that there is still some ambiguity in the pose of the endogeneous MQ. However, please note that the choice of pose for MQ does not affect any of the arguments made in the manuscript.

There are many geometry violations in the isoprenoid chains of the modeled menaquinones (see chain F MQ9 607 C34 for an extreme example), the authors should double check their ligand restraints file. Overall the density for chain F MQ9 607 is unconvincing that it belongs to a menaquinone. The density for the isoprenoid tail for chain F MQ9 609 is very weak and difficult to follow why it was traced in the way it was (regions of it are not in density, whereas significant density in the same vicinity remains unmodeled). This issue is also seen in other regions of the map/model, i.e. there are several clear densities for lipids that are not modeled and modeled acyl chains that are not in clear density. The authors should carefully reevaluate their lipid and menaquinone modeling before publication.

We thank the reviewer for their attention to detail and apologize for this oversight. Many of the lipids and MQ models were “inherited” from the model from Gong *et al.*, which we used to start building our model, as described in the methods section, and these poorly modelled features were not subjected to sufficient scrutiny. MQ9 609 has now been removed and the isoprenoid tail of MQ9 607 has been altered to fit the density originally occupied by the isoprenoid tail of MQ9 609. Lipids throughout the complex (9XX 302, 9XX 204, 9Y0 201, 9YF 504, and 9YF 501) have been altered to better fit the density and the models have been reprocessed in phenix with new restraints (new report included).

The authors first mention His368 on page 6 but do not discuss its significance until pages 7 and 8. Clarity could be improved by discussing the significance of His368 when it is first mentioned.

We thank the reviewer for this suggestion. To ‘prime’ the reader for the proposed role of His368, we have added the following sentence to immediately after where His368 is mentioned for the first time (p. 6, 4^th^ paragraph):

“His368 from QcrA is believed to have an important role in CIII, accepting a proton during quinone oxidation at the Q_P_ site (Mulkidjanian, 2005).”

His368 is then mentioned again on the 2^nd^ paragraph of page 7:

“In this position, the naphthoquinone head group is ~14 Å away from the FeS cluster and the hydroxyl proton is ~15 Å from His368, which is too far for rapid coupled electron and proton transfer from MQH_2_ to FeS and His368, respectively.”

With the main discussion of His368 at the end of page 7 and beginning of page 8.

Given that the co-purified menaquinone seen in the inhibitor-free enzyme is likely oxidized, wouldn't the Q1b position more likely be for oxidized menaquinone as opposed to the reduced menaquinol as depicted in Figure 4A? If so wouldn't the menaquinone in the Q1b position likely be more representative "exiting" the QP-site after donating its electrons, as opposed to a menaquinol "entering" the QP-site in order to donate its electrons? If so, what would the implications be for the reversibility of the Q-cycle and semiquinone formation at the QP-site, to have a menaquinone binding site adjacent to the QP site that is too distance to donate electrons to or accept electrons from the FeS cluster or heme bL?

We thank the reviewer for this excellent suggestion. We agree that the co-purified endogenous menaquinone is most likely oxidized. Hence, the cryoEM structure almost certainly shows the position of the oxidized menaquinone. The Q1b position is along the only entry/exit path to/from the Q1a position where menaquinol oxidation can take place. Therefore, we speculate that during turnover, when the menaquinone pool is mostly reduced, MQH_2_ would have to access the Q1b position, at least transiently, because this position is found along the MQH_2_ transfer trajectory. We have updated the text and figure to show the model where reduced MQH­_2_ accesses the Q1b position, moves to the Q1a position to transfer its electron to the FeS group, and then returns to the Q1b position (p. 9, 2^nd^ paragraph):

“Although the endogenous MQ found in the structure is most likely oxidized, the position it occupies is along the access path to the FeS center. We speculate that reduced MQH_2_ also adopts the same position, at least transiently, during turnover. This binding site within Q_P_ is known asthe Q1b position. As discussed above, the Q1b position is too far for rapid transfer of protons and electrons from MQH_2_ to His368 and the FeS center, respectively (Figure 4A). In contrast, movement of MQH_2_ deeper within the Q_P_ site to the Q1a position would bring it less than 10 Å from FeS, close enough to donate electrons to the redox center and protons to His368 (Figure 4B). The oxidized MQ could then return to the Q1b position, as observed in the present structure (Figure 4C), before it is exchanged for MQH_2_ from the membrane. In an emerging model for CIII_2_CIV_2_ function, the Q1b position serves as a "stand-by" site for MQH_2_, with oxidation of the substrate occurring only upon relocation to Q1a (see (Moe et al., 2021; Mulkidjanian, 2005)). The structure of CIII_2_CIV_2_ with telacebec bound shows that the compound serves as dual site inhibitor (Figure 4D), with the imidazopyridine group bound to the Q1a position and A-phenyl portion of the tail bound to the Q1b position. Indeed, the possible hydrogen bond between Thr308 and the carboxyl group in the linker region of telacebec could also occur between Thr308 and MQ, stabilizing it in the Q1b position. Therefore, the observed pose of the inhibitor within the enzyme not only blocks MQ access to the FeS center but fills the Q_P_ site entirely.”

We have updated Figure 4 to show both the reduced and oxidized forms of MQ accessing the Q1b position.

Reviewer #2 (Recommendations for the authors):My only recommendation is using multiple biological replicates (at least two, better three) for the activity assays.

We apologize for not making clear that the measurement were indeed from multiple samples of purified proteins. However, it is important to note that the repeated assays are not simple technical replicates (e.g. multiple instrumental measurements made on the same sample). Each oxygraph measurement is done one-at-a-time in series, requiring creation of a new assay mixture with multiple reagents, reassembly of the reaction chamber, equilibration, and independent measurements of background oxygen consumption rates. Thus, protein batch is not the only significant source of variance in assays, particularly in inhibitor titrations where activity is expressed as a fraction of the uninhibited enzyme.

We have modified the relevant text (p. 6, 1^st^ paragraph):

“With 500 nM SOD added, CIII_2_CIV_2_’s DMWH_2_:O_2_ oxidoreductase activity was measured at 91 ±4 e^-^/s (± s.d., n=6 independent assays with three each from two separate batches of protein), which is nearly an order of magnitude greater than the apparent activity found previously (Wiseman et al., 2018).”

And (p. 6, 2^nd^ paragraph)

“Titrations of CIII_2_CIV_2_ activity with varying concentrations of telacebec (Figure 2C) show an IC_50_ of 53 nM ±19 (± s.d., n=3 independent titrations, with two titrations from one batch of purified protein and a third titration from a second batch of purified protein) with 65 nM CIII_2_CIV_2_ and 100 µM DMW.”

Reviewer #3 (Recommendations for the authors):Specific points to address:1. Readers will be interested to see how reported mutations in qcrB in M. tuberculosis that confer resistance to Q203 map onto the CIII2CIV2 inhibited complex. How does the H190Y mutation reported for TB47 map to the new structural data with the Q203-inhibited structure (see Lu et al., ACS Infectious Diseases 5:239-249 (2019) PMID: 30485737).

We have added the requested information to the paragraph following the description of the telecebec binding pose as well as an additional panel in the Figure 3 —figure supplement 1 (p, 8, 4^th^ paragraph):

“The structure also suggests how mutations can provide resistance to telacebec and why telacebec selectively inhibits mycobacterial CIII_2_. The mutation T313A in *M. tuberculosis* (T308A in *M. smegmatis*) confers resistance to telacebec (Pethe et al., 2013), likely by removing the stabilizing hydrogen bond with the carbonyl group from the linker region of the inhibitor proposed above (Figure 3C and D). *M. smegmatis* grown in the presence of the telacebec analogue TB47 developed the mutation H190Y (Lu et al., 2019), which is adjacent to the cd1 helix and may alter the shape of the Q_P_ binding site (Figure 3 —figure supplement 1C).”

2. A short paragraph might be appropriate on how the new structural data reported herein could be used to improve on inhibitors of this supercomplex.

This information has now been added into the paragraph following the description of the telacebec binding pose (p. 8, 3^rd^ paragraph):

“With telacebec and congeners having nanomolar inhibitory activity for both CIII_2_CIV_2_ and *M. tuberculosis* growth in vitro, antimycobacterial activity need not be improved for therapeutic purposes. However, the structural analysis reported here provides constraints and minimal requirements for activity of imidazopyridines and isosteric heterocycles with improved pharmacokinetic and physicochemical properties. Optimized physicochemical properties are important for drug production, including synthesis and purification. Improved pharmacokinetic properties could be enabled by design of analogues that retain target activity but are not recognized by mycobacterial efflux pumps, which are known to remove telacebec from bacterial cells to attenuate its antimycobacterial activity (Jang et al., 2017).”

3. A structural evaluation of why Q203 is unable to inhibit the mitochondrial Complex III structure could be included.

We have added the following analysis (p. 8, 4^th^ paragraph):

“The structure also suggests how mutations can provide resistance to telacebec and why telacebec selectively inhibits mycobacterial CIII_2_.”

“The selectivity of telacebec for mycobacterial CIII_2_CIV_2_ may derive, in part, from the lack of Thr308 in mammalian mitochondrial CIII_2_ (Figure 3 —figure supplement 1D). In addition, there may be clashes between the rigid telacebec tail and both Leu150 and Ile146 in mammalian CIII_2_ (bovine numbering) due to the different location of the cd1 helix and the bulkier side chains in this region of the mammalian protein (Figure 3 —figure supplement 1D). Interestingly, the mutation I147F (*M. smegmatis* numbering) results in resistance to the inhibitor stigmatellin in the *Saccharomyces cerevisiae* CIII_2_ (di Rago et al., 1989)”

4. The Beites et al., 2019 reference should also include these two citations:Lee BS, et al., EMBO Molecular Medicine 13: e13207 (2021)Kalia, NP, et al., Proceedings of the National Academy of Sciences of the United States of America 114:7426-7431 (2017)

We thank the reviewer for pointing out these references and have added them in the final paragraph where we cite Beites *et al.,* (2019) regarding dual inhibition of cyt. *bd* and cyt*. bcc*-*aa3*.